# Signal-based optical map alignment

**Mehmet Akdel**[1], **Henri van de Geest**[2¤a], **Elio Schijlen**[2], **Irma M. H. van Rijswijck**[3], **Eddy J. Smid**[3], **Gabino Sanchez-Perez**[1,2¤b], **Dick de Ridder**[1]*

**1** Bioinformatics Group, Wageningen University, Wageningen, The Netherlands, **2** Applied Bioinformatics, Bioscience, Wageningen Plant Research, Wageningen, The Netherlands, **3** Laboratory of Food Microbiology, Wageningen University, Wageningen, The Netherlands

¤a Current address: Genetwister Technologies BV, Wageningen, The Netherlands
¤b Current address: Hudson River Biotechnology, Wageningen, The Netherlands
* dick.deridder@wur.nl

**Data Availability Statement:** All relevant data and code are available at https://github.com/akdel/OptiScan and https://github.com/akdel/OptiMap.

**Funding:** This work is part of the Open Technology Programme with project number 14516, (partly)

## Abstract

In genomics, optical mapping technology provides long-range contiguity information to improve genome sequence assemblies and detect structural variation. Originally a laborious manual process, Bionano Genomics platforms now offer high-throughput, automated optical mapping based on chips packed with nanochannels through which unwound DNA is guided and the fluorescent DNA backbone and specific restriction sites are recorded. Although the raw image data obtained is of high quality, the processing and assembly software accompanying the platforms is closed source and does not seem to make full use of data, labeling approximately half of the measured signals as unusable. Here we introduce two new software tools, independent of Bionano Genomics software, to extract and process molecules from raw images (OptiScan) and to perform molecule-to-molecule and molecule-to-reference alignments using a novel signal-based approach (OptiMap). We demonstrate that the molecules detected by OptiScan can yield better assemblies, and that the approach taken by OptiMap results in higher use of molecules from the raw data. These tools lay the foundation for a suite of open-source methods to process and analyze high-throughput optical mapping data. The Python implementations of the OptiTools are publicly available through http://www.bif.wur.nl/.

## Introduction

The last decade has seen a sharp rise in the number of available genome sequences, caused by the ubiquity of second and third generation sequencing technologies [1]. However, the quality of such genomes is not always consistent and is highly dependent on the genomic features of the organism. While short, feature-rich genomes (e.g. those of bacteria) are now routinely assembled into a single chromosome, complex genomes such as those of fungi and plants—which can be large, repetitive, sometimes polyploid or even aneuploid—still require a combination of sequencing technologies and assembly and scaffolding methods. As a result, many eukaryotic genome sequences are actually draft genomes, as they are not fully assembled into structurally coherent chromosomes. This hampers the use of these genomes in genome-scale haplotyping and genotyping-by-sequencing, in -omics studies to reveal phenotypic effects of

financed by the Netherlands Organisation for Scientific Research (NWO), domain Applied and Engineering Sciences, and supported by Bayer CropScience NV, Genetwister Technologies BV, Rijk Zwaan BV and SESVanderHave NV. Yeast optical map data was generated in a ZonMW Enabling Technologies Hotels project (435002001). The funders had no role in study design, data collection and analysis, decision to publish, or preparation of the manuscript.

**Competing interests:** No authors have competing interests.

large-scale copy number variation and structural variation, and in fundamental research on the molecular mechanisms of genome evolution.

The difficulty in assembling complex genomes is caused to a large extent by repetitive sequences, i.e. low complexity repeats, tandem duplicated genes and gene clusters, which exceed the longest possible read length allowed by the sequencing technology used [2]. Long read sequencing as offered by PacBio and Oxford Nanopore promises to help resolve repetitive genomes, but the current average read lengths of 10–100kb are insufficient to bridge the long repeat regions common in higher eukaryotic genomes [3, 4] and the technology is not yet cost effective in detecting structural variation across large numbers of genomes. Currently, the sole available technology that overcomes these issues is high throughput optical mapping [5–7].

Optical mapping (OM), as introduced nearly 30 years ago, involves stretching out and immobilizing (long parts of) individual chromosomes on a glass slide, cutting the DNA with restriction enzymes, staining and then imaging the fragments by fluorescence microscopy [8]. The process has previously been used to help compose complex genomes, but was costly, intricate, error-prone and low-throughput. It has therefore remained underused, despite its potential to resolve ambiguities in the many draft genomes that have recently become available [9]. This changed with the introduction of high-throughput OM technology [10], currently led by Bionano Genomics' (BNG) Saphyr DNA nanochannel platform [11]. High molecular weight DNA is labeled at specific motif sites without double-stranded cleavage and run through the nanochannels to obtain fluorescence images. These images are then processed into label coordinates per DNA molecule, typically ranging from 100kb up to several Mb. This data can then either be used to scaffold existing sequence-based assemblies [12–15], or directly assembled into consensus optical maps which can help detect structural variation [16–19].

The analysis of OM data produced by BNG platforms has two main challenges. First, due to a relatively high error rate in the labelling process, substantial numbers of false negative and false positive labels are found (10–20%). Second, the flexibility of DNA molecules leads to stretching, due to which the distance between the same sites may vary between different OM molecules [20]. A number of optical mapping alignment and hybrid-scaffolding methods have been proposed, initially targeting previous generation, low-throughput optical mapping data [21–23]; for an overview, see [24]. However, these tools are not scalable to BNG optical map data; instead, RefAligner, the software bundled with the BNG platform, is often used for the alignment and assembly of such data.

Before the alignment process, the signals from raw OM molecules are translated into a sequence of distances between label sites. RefAligner uses an approximate alignment method on these sequences. Molecule-versus-molecule alignments obtained with this algorithm form the basis of *de novo* optical map assembly and hybrid-scaffolding. While RefAligner is fast and yields decent results, it seems inefficient in that around half of the molecules remain unused; moreover, it is closed-source and thus prohibits inspection or improvement. A number of alternative alignment methods have been proposed for high-throughput data [25–27]. However, these methods were developed with molecule-to-reference genome alignments in mind, not molecule-to-molecule alignments. In addition, similar to RefAligner, all of the current OM aligners discard information on the intensity of label sites and only make use of their locations.

Here we introduce OptiScan, a tool to extract optical maps as one-dimensional signals of label site intensities from raw image files. OptiScan can convert these signals into the BNX format, to be used in BNG assembly, and improves assembly performance. A second tool, Opti-Map, uses a digital signal processing approach to align the optical map signals found by OptiScan. Based on Irys optical mapping datasets of yeast and eggplant, we demonstrate that OptiMap outperforms the state-of-the-art software, and more accurately aligns more unique molecules.

## Materials and methods

### The BNG optical mapping platform

Unlike traditional optical mapping methodology, the BNG approach does not digest DNA molecules but fluorescently labels them at specific sequence sites (Fig 1). A second, aspecific label is applied to the DNA backbone for overall detection of molecules. These two different labels emit different wavelengths of light when excited. Labelled DNA molecules are loaded into BNG flowcells, which contain chambers to unwind the DNA followed by (tens of) thousands of nanochannels, each wide enough for a single DNA molecule to be pulled through. The loading process is regulated by an electrical current across the flowcell, which in effect draws the charged DNA molecules in. In time, the DNA molecules start extending to their full length into the nanochannel. When a sufficient number of molecules are extended in the channels, the chip is scanned and the instance is recorded. This process is repeated, each time pulling in new molecules. Each chip can be used for a finite number of scans before the channels get clogged. Several such runs of the same sample can then be combined (if needed, depending on genome size), with each run contributing several tens to hundreds of thousands of molecules.

### OptiTools

The two tools we describe here, OptiTools, offer an alternative to the image detection and alignment tools provided by BNG. Fig 2 gives an overview of the proposed workflow. OptiScan detects molecules and provides these to OptiMap for subsequent alignment. Alternatively, the molecules can be exported in the BNG proprietary BNX file format, containing molecule starts and ends, label positions and summaries of the peaks detected, for further processing by BNG software. OptiScan and OptiMap are written in Python version 3.6 (Python Software Foundation, https://www.python.org/) using a number of packages.

### OptiScan

The output of the Irys platform consists of 2 sets of paired, raw 512 × 512 pixel 16 bit TIFF images: backbone frames and label frames (Fig 3), i.e. fluorescent images of the entire labeled

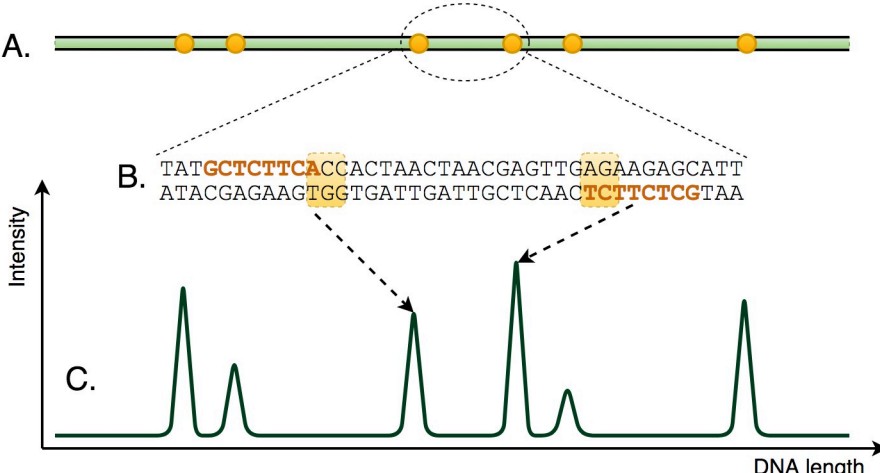

**Fig 1. The principle of BNG optical mapping.** Long DNA molecules (A) are fluorescently labelled at specific sites (B). Signals are then captured (C) in which peaks correspond to these sites.

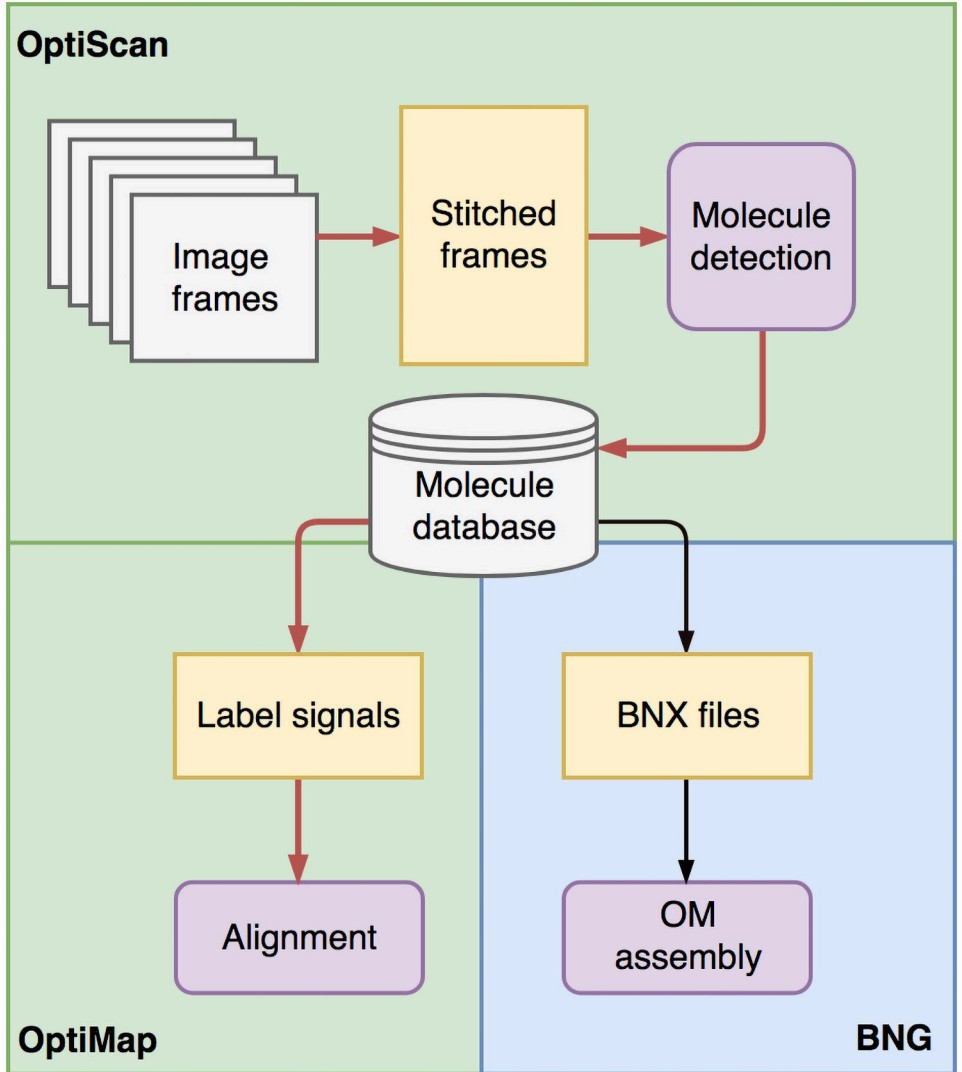

**Fig 2. OptiTools workflow.** OptiScan detects molecules and stores these in a database, which can then be used by OptiMap for molecule-to-molecule or molecule-to-reference alignment, or exported for use with BNG methods in a BNX molecule file.

DNA molecule and the individual target sites, respectively. Each frame pair corresponds to the same location on the chip. A set of frames together makes up a single scan of a chip, for example 12 frames high and 94 frames wide (depending on the flow-cell version). The Saphyr platform outputs the backbone and label images as columns, of varying numbers, each containing six concatenated 2048 × 2048 pixel image frames.

**Frame stitching.** As DNA molecules can span multiple frames, it is necessary to stitch each column of frames into composite columns. For Irys data, the image frames can be grouped by columns to continue with the compositing step. Saphyr columns are already encoded in single image files; therefore, we first split these columns into corresponding frames. The stitching process in OptiScan involves computation of various transformations including rotation, scaling and translation to optimize the overlap between adjacent frames. OptiScan uses the following steps to convert raw images into aligned, paired columns of backbone and label images (illustrated in Fig 4):

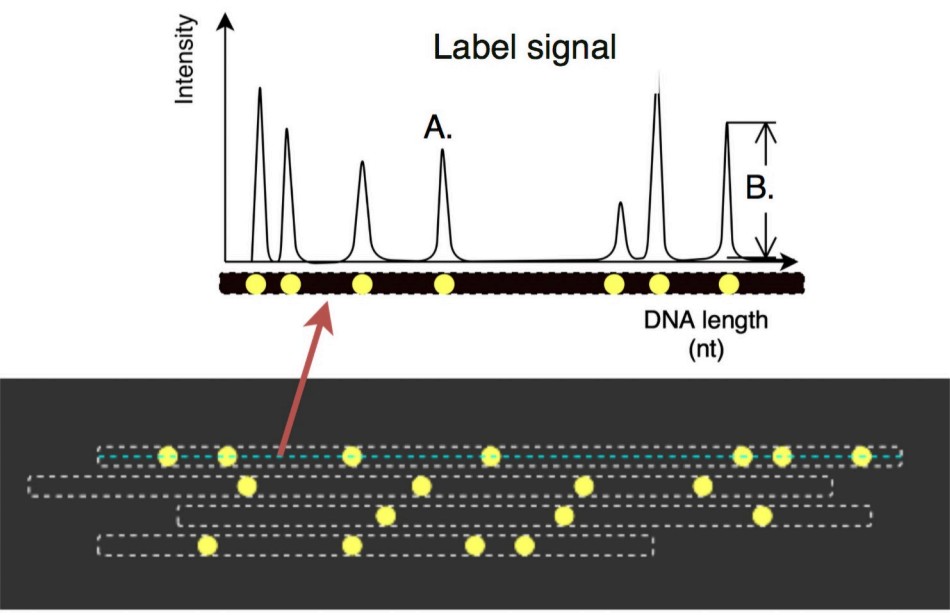

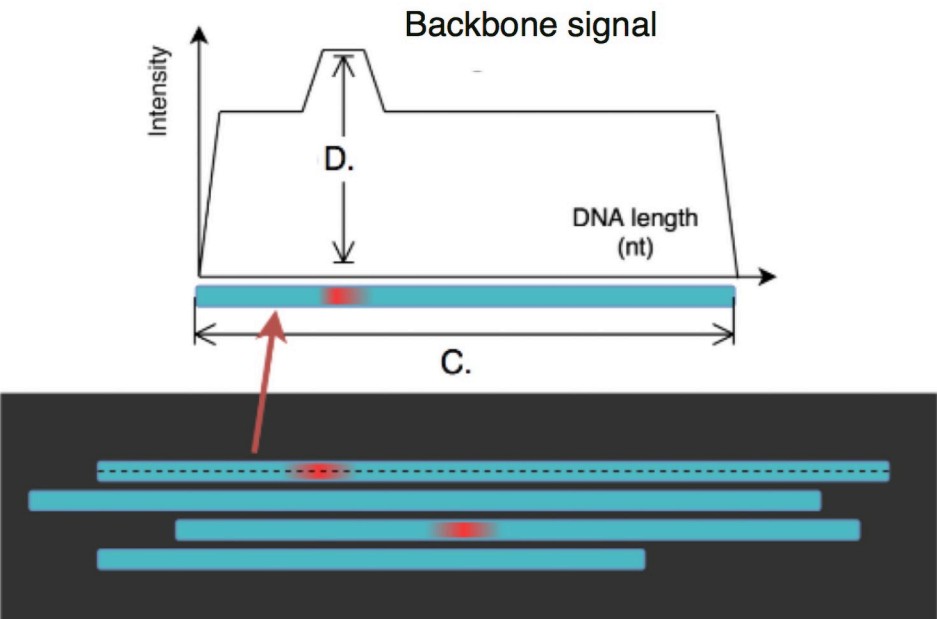

**Fig 3. Frame pairs as produced by the BNG platform.** Each scan produces two frames (images), with labels (top) and backbones (bottom). The top molecule's label and backbone intensities are illustrated as 1D signals above the frames. In the label frames, peaks correspond to label centers (A), with peak heights indicating label intensity (B). In the backbone frames, stretches of equal intensity delineate the molecules (C) with occasional higher intensity stretches indicating possible DNA entanglement (D). Note that the frames shown here are in a horizontal orientation for illustration purposes.

1. To facilitate molecule detection, gradients in the background intensity are removed by applying a tophat filter, using a circular kernel with a diameter of one pixel more than the average molecule width (11 pixels) [28].

2. As not all frames are captured in a perfectly vertical orientation, backbone labels are aligned to vertical by applying a slight rotation to each frame. To optimize the rotation angle, the

**Fig 4. Frame stitching.** A simplified illustration of rotation (A), horizontal (B) and vertical (C) translation towards the stitched frames (D).

sum of projected pixel intensities is calculated for each column in the rotated frame for angles ranging between −1deg and 1deg with 0.01deg interval. The angle which yields the maximum sum in any column is then selected. The same rotation is also applied to the paired label frame.

3. Next, backbone frames are stitched together column by column based on the small overlap between frames. Keeping the first frame fixed, subsequent frames are progressively shifted to align frame-spanning molecules. We computed two dimensional cross-correlation (implemented using the scipy library "fftpack.correlate" function [29]) between adjacent frames and applied the shift resulting in the highest cross-correlation value. As in the rotation step, an identical translation is then applied to the paired label frame.

4. To remove small size differences between backbone and label frames, each label frame is resized by using a scaling factor within a ±0.01% range in steps of 0.001, and the factor which results in the maximum cross-correlation between the resized label and backbone frames is applied.

**Molecule detection.** A fundamental step in the processing of BNG optical map data is to define and extract individual DNA molecules from the stitched images. The molecule detection algorithm in OptiScan consists of two steps. In the first step, molecule boundaries are detected in the backbone images. To this end, each image is convolved with a vertical edge detection kernel, $[1|1|1|1|1|1]^T \times [1|-1]$, followed by a simple intensity threshold (default setting 110) to detect right-edge molecule boundaries. These boundaries are collected for individual molecules using the SciPy functions `label()` and `find_objects()`, yielding starting points for more accurate molecule detection.

The second step involves extracting the target labels which fall within each of the determined molecule boundaries. Backbone center lines are detected as peaks in the three pixels immediately to the left of the molecule boundaries. Intensity peaks are then detected as local maxima at the corresponding positions in label frames on these backbone center lines. This yields one-dimensional intensity signals of molecule backbone and target site labels. The latter present themselves as a series of peaks where each peak corresponds to a target site (Fig 3).

The resulting measurements are stored in a combination of an SQLite database and memory mapped files. For each molecule, the SQLite database records its ID, the column in which it was detected, its coordinates in that column, molecule length, average intensity of backbone

and labels, and this information is linked to the memory mapped files which store the raw backbone and label signals.

**Molecule quality analysis and BNG assembly.** To allow further processing by BNG software, the detected molecules can be exported into BNG's BNX file format. BNX files store sets of molecule and label locations, as well as label peak signal-to-noise ratios (SNRs). These SNR values are used as an indication of label peak quality, to filter out poor labels before assembly. For export, we transformed our label signals into coordinate-SNR pairs by detecting peak locations and converting these from pixels into DNA base pair (bp) units. We estimated the SNR for each peak by dividing the peak intensity by the background intensity.

For subsequent assembly, the BNG software requires a minimum SNR threshold to be set. To allow a fair comparison to molecules found by the BNG toolset, we first used the Molecule Quality Report (MQR) tool from BNG to choose an optimum SNR value for each dataset. This tool maps a randomly selected set of 50,000 molecules onto an *in silico* digested reference genome and returns the number of OM molecules mapped. We then found the SNR threshold as the one yielding the highest mapping rate, independently for molecules generated by OptiScan and by the BNG software. The optimum thresholds were 3 for the BNG molecules and 2.8 for the OptiScan molecules, resulting in a map rate of approximately 31% for both.

We assembled both sets of molecules *de novo* using different numbers of scans—5, 10 and 30—to study the influence of coverage, and calculated assembly confidence values using the BNG toolset. These values are found by aligning the assembled contigs to the *in silico* digested reference genome and taking the $-\log_{10}$ of the alignment *p*-value.

## OptiMap

**Molecule-to-molecule alignment.** Current methods for optical map alignment are generally based on representing fragments as a sequence of label interval lengths, followed by sequence-based alignments, often using a form of dynamic programming [25–27]. To allow for the high false positive/negative label rates, the alignment function incorporates insertions and deletions; to deal with molecule stretching, the interval lengths can differ to some extent, as captured in some alignment goodness metric [22]. However, these procedures discard peak height and SNR information.

Here we take a different approach. Peak heights were observed to be informative, in that relative heights are reproduced in different molecules (see Results, "Label intensities are informative"). To capture this, we align signals directly, without translation to sequences, using cross-correlation. Cross-correlation is a digital signal processing (DSP) function based on convolution, commonly used for assessing similarity between two signals. The position at which the cross-correlation function is maximal corresponds to the shift with which the two signals align best, and the value of the function at this position can serve as an alignment score. Based on this approach, we have three modes of computing an alignment with varying degrees of precision/recall trade-offs, as described below.

**Naive mode.** Naive mode makes use of the cross-correlation method with a minimum overlap threshold, which is set to 125 Kb (250 pixels) by default. Without normalization, parts of signals with higher label densities lead to high scores irrespective of the specific label pattern. To prevent this, the cross-correlation score is normalized by dividing by the sum of squared differences between the signals in the overlap. Many molecules contain some high amplitude labels, which have a detrimental effect on the alignment score. In order to avoid false negative alignments, we log-transformed the raw signals prior to the normalized-cross correlation procedure (Fig 5B). As the orientation of molecules is unknown, alignments are performed using both the original and reverse orientation of the shorter signal and the maximum normalized

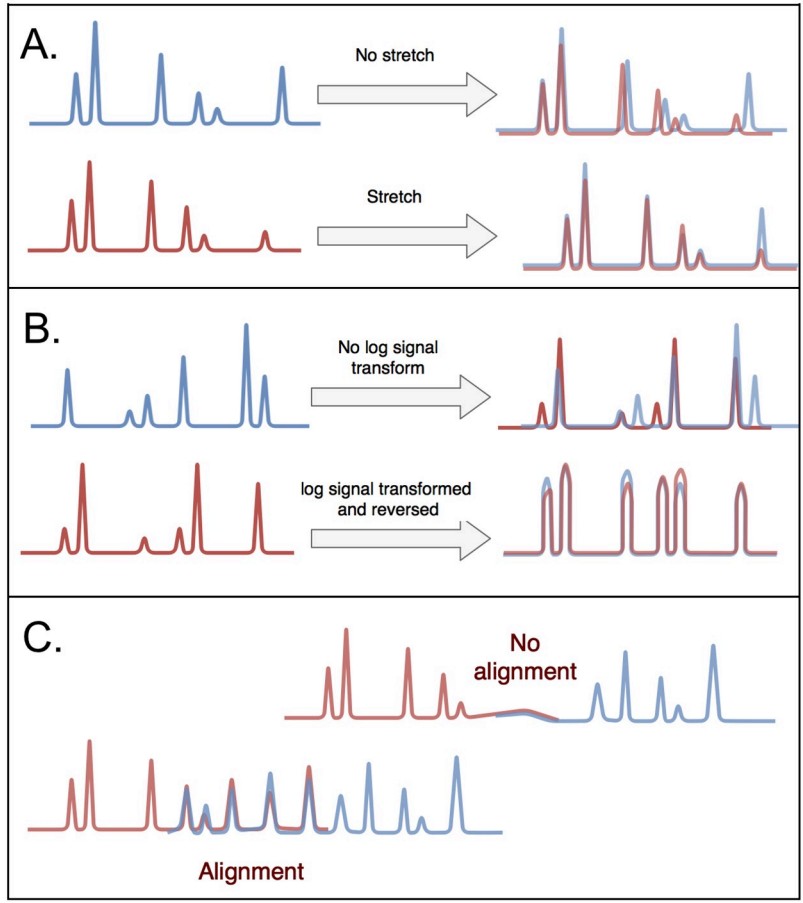

**Fig 5. The OptiMap alignment procedure. A**. Allowing for molecule stretching.**B**. Log-transformed signal correlation scores help confirm alignments based on raw signals. **C**. Requiring a minimum number of overlapping labels.

score over these alignments is assigned as the correct orientation. The outcome of an alignment is an optimum overlap range with an identity score $S$ between two overlapping signals $m$ and $n$, which ranges between 0 (no similarity) and 1 (identical):

$$S(m, n) = \max_{\tau} \frac{\sum\limits_{i=\max(1-\tau,1):\min(L_m-\tau-1,L_m),j=\max(1+\tau,1):\min(L_n+\tau+1,L_n)} m_i \cdot n_j}{\sqrt{\left(\sum\limits_{i=\max(1-\tau,1):\min(L_m-\tau-1,L_m)} m_i \cdot m_i\right) \cdot \left(\sum\limits_{j=\max(1+\tau,1):\min(L_n+\tau+1,L_n)} n_j \cdot n_j\right)}}$$

where $m$ and $n$ are zero-padded vectors of length $L_m$ and $L_n$ respectively and $\tau$ is restricted to a range such that the minimum overlap is 125 Kb.

We designate a pair of molecules as an aligning pair if the score $S$ is above the threshold (default: $S > 0.60$) and discard the remaining pairs. A limitation of this naive alignment approach arises from molecules that contain long stretches with few labels, resulting in unspecific label patterns. These unspecific stretches can match with high scores, leading to false positive alignments. The second limitation of such an approach is due to molecule stretching. This phenomenon can lead to both local and global stretching of molecules, which lowers the alignment scores, leading to false negative alignments. The following two modes extend the naive alignment approach based on prior knowledge of OM molecule signals and data coverage.

**Sparse overlap mode.** To deal with very low coverage datasets, we also developed a sparse overlap mode in which we aim to remove false positive alignments (unspecific alignments) and capture missed alignments from stretching by making use of specific properties of OM molecules:

1. To avoid false negatives caused by molecule stretching, the shorter of the aligning signals is resized by re-sampling within a range from -5% to +5% of its length, in steps of 1% and aligned again. The maximum normalized score over these alignments is then used (Fig 5A).

2. To prevent cases where noisy molecule-ends without labels are aligned (false positives), alignments based on less than 9 labels (per molecule) in the overlapping region are discarded (Fig 5C).

Since this mode adds further computational complexity, it is not viable for compute large numbers of alignments. Therefore, we first use the naive mode with a lower score threshold than default ($S > 0.55$) to compute pairs of alignment candidates. These are subsequently subjected to realignment in sparse mode.

**Dense overlap mode.** OM data is particularly useful at higher coverages, as most subsequent analyses depend on a *de novo* assembly step, in which genome map contigs are constructed with a consensus pattern of labels averaged from all molecules used. High coverage data implies that each molecule overlaps with multiple others, knowledge which can be used to increase alignment precision. Our three step approach first involves the use of naive mode with a relatively high score threshold of $S > 0.65$, to avoid finding false positive alignments. In the second step, the alignment pairs found are used to create an undirected graph on which transitive closure is performed, linking molecules which are not found to align directly but have neighbors which align. These are most likely false negatives left over from the first step. In the last step, we perform another round of sparse alignment including the newly found pairs from transitive closure, this time using a lower score threshold than the default ($S > 0.55$).

**Molecule-to-reference alignment.** To map molecules to a reference genome, we first perform *in silico* digestion to find positions where labels can be found. We then simulate a reference square-wave signal using these positions and align molecule signals to it. Each peak (with arbitrary amplitude) is $p$ units wide. Unit length $l$ and peak width $p$ were set based on our previous analysis of raw optical map signals and on BNG imaging specifications [30] as $l = 500$bp and $p = 10$. We then simulate square-wave counterparts of the real molecules by using the detected label sites in each molecule. Finally, the molecule-to-reference alignment is performed using either the naive or sparse approached explained in the section above (Molecule-to-molecule alignment) to align each molecule to the reference signal.

## Dataset

We demonstrate OptiScan and OptiMap on data obtained on yeast, from the reference *Saccharomyces cerevisiae* strain (S288C, [31]), and on eggplant, *Solanum melongena*. For both organisms protoplasts were first formed and embedded in low melting temperature agarose plugs, which were subsequently treated with protK, RNAse and gelase to release high molecular weight DNA. This DNA was used for nicking by Nt.BspQ1 (NEB), labelling, repair and staining according to BNG's protocol. A single run (30 scans) of optical mapping was then performed on the BNG Irys platform, resulting in dataset of 30 BNX files, storing the label coordinates of each of the detected molecules and labels, as well as the original raw image data with accompanying meta-data regarding the dimensions of the chip. For *Saccharomyces cerevisiae*, the reference genome sequence was obtained from SGD [32] (https://www.yeastgenome.org/, release

R64–2-1). For *Solanum melongena*, we used a previously assembled, high coverage optical genome map for the purpose of alignment and assembly quality assessments.

## Results and discussion

### OptiScan and BNG detect different molecules

We first applied OptiScan to retrieve molecule signals from the yeast optical mapping images. Visual inspection confirmed the contiguity of molecules which extended through multiple frames. For this data set, OptiScan yielded an average of 7,034 molecules per scan, with a minimum length filter of 250 pixels. In comparison, the BNG software detected 6,386 such molecules on average. One-to-one molecule position comparison between BNG and OptiScan showed that 8% of the molecules detected differed by 20% or more in molecule length. We visually inspected these differing molecule boundaries and found that OptiScan has a tendency to split molecules at low intensity locations, as illustrated in Fig 6. This is safer, as ignoring such stretches may lead to the inclusion of chimeric signals, combining two molecules that happen to be close together in the nanochannel. Such chimera complicate the assembly process.

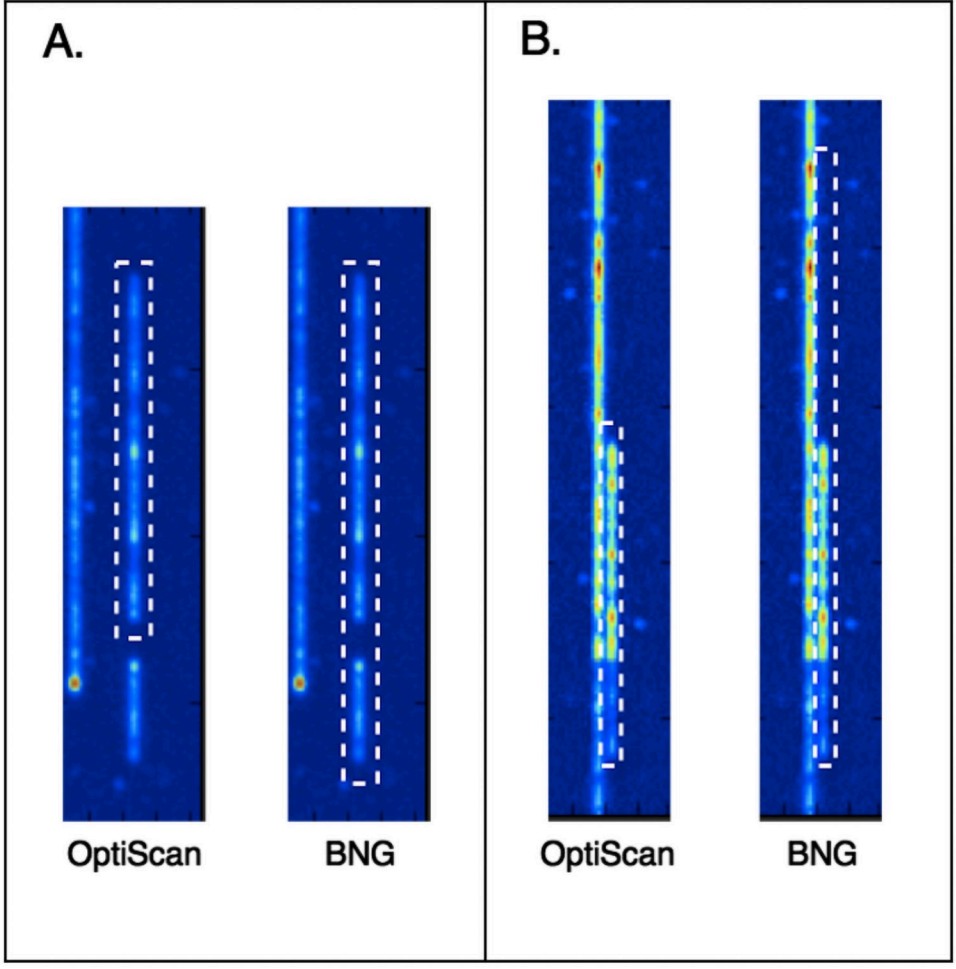

**Fig 6. Molecule detection (yeast) by OptiScan and BNG.** Two examples of cases where OptiScan errs on the side of caution in molecule detection. In both, the BNG molecule detection routine seems to generate erroneous molecules.

**Table 1. BNG assembly pipeline results using both OptiScan (O) and BNG (B) molecules of yeast.**

| Cov. | | Aligned length (Mb) | Labels | TP | FP | FN | Avg. conf. | Total conf. |
|---|---|---|---|---|---|---|---|---|
| 30 | O | 11.51 | 1653 | 1539 | 114 | 15 | 100.21 | 2104490 |
| | B | 11.26 | 1519 | 1414 | 105 | 13 | 95.16 | 1903310 |
| 10 | O | 10.90 | 1560 | 1496 | 64 | 2 | 84.13 | 2019333 |
| | B | 10.89 | 1450 | 1389 | 61 | 8 | 74.10 | 1852692 |
| 5 | O | 7.78 | 1147 | 1103 | 44 | 1 | 55.68 | 1447870 |
| | B | 7.54 | 1042 | 1008 | 34 | 4 | 49.76 | 1293859 |

Cov.: coverage. TP, FP, FN: true positive, false positive and false negative labels. Conf.: confidence scores (higher is better) as calculated by the BNG software.

## OptiScan improves BNG assemblies

Although OptiScan molecules appear to be of somewhat higher quality, it is unclear to what extent these differences in molecule detection actually affect assembly. To assess this, we compared assemblies made using the BNG software based on both BNG-detected molecules and OptiScan-detected molecules for different coverage levels, and calculated assembly confidence values using the BNG toolset (see Materials and methods, "Molecule quality analysis and BNG assembly"). Quantitative results are presented in Table 1 and some examples of assembled contigs are shown in Fig 7.

OptiScan provides some improvements over BNG molecule detection. First, the number of labels found in OptiScan molecules is generally higher than in BNG molecules (Table 1). While the number of false positives and false negatives is slightly higher as well, the majority of the additional labels are correct, which is also reflected in the higher confidence values. This could be due to a better resolution of nearby label sites (Fig 7A), indicating a better performance of the OptiScan peak finding method. While labels on the OptiScan-based contig match the reference labels well, in the BNG contig a single label matches multiple adjacent reference labels. OptiScan also showed improved contiguity of contigs assigned to the same chromosome, an example of which is shown in Fig 7B.

A higher label resolution not only helps obtain better assemblies, but can also result in more accurate haplotypes and short-range structural variations. Increased resolution may also explain why a long repeat, found to map on chromosome XII of the yeast genome in a region containing ribosomal RNA and retrotransposons, had different lengths using the two types of data (Fig 7C; note that the correct number of repeats is unknown, as this repeat is not a part of the reference).

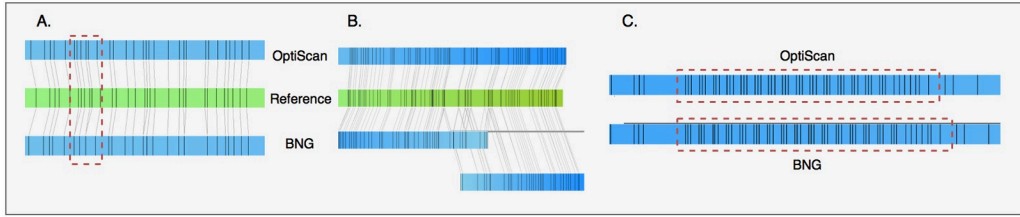

**Fig 7. Comparison of 3 different contigs assembled by the BNG assembler using molecules extracted by OptiScan and by BNG software.** A. OptiScan molecules can have higher resolution, which often more accurately matches the reference genome (i.e. the *in silico* generated optical map). B. In some cases, OptiScan-based assembly results in better contiguity. C. A long repeat on chromosome 12, not present in the reference genome, was assembled differently based on the two molecule sets.

The results also demonstrate that sufficient coverage is essential, but for this dataset assembly quality at 10x coverage is already close to that at 30x.

## Label intensities are informative

While existing approaches to molecule alignment and assembly only take inter-label distance into account, inspecting OptiScan's results indicated that there is also information in relative peak heights (in BNG terminology, the signal-to-noise ratio or SNR). That is, label intensities seemed to consistently vary between genomic positions, likely due to local sequence environment. To verify the extent of this phenomenon, we took all label positions in the assembly and extracted the original label SNRs from the molecules used in the consensus assembly. We found that at a large number of label positions, the average log(SNR) deviates from the overall average; at 25% of the positions, significantly so (two-sided $t$-test, $p < 10^{-2}$). This confirms a pattern of consistent variation in relative peak heights over different genomic positions, which we next set out to exploit in a signal-based alignment approach. Peak patterns could be further explored in the future to obtain more insights into their meaning.

## OptiMap aligns more molecules with higher precision

As implied earlier, one of the shortcomings of RefAligner is its seeming inefficiency in terms of the fraction of molecules and alignments used from original data. Here, we show that OptiMap's molecule to molecule alignment finds and uses a larger number of unique molecules and alignments.

**Compilation of molecule sets with ground truth alignments.** We assessed and compared the alignment performance of OptiMap and RefAligner based on both yeast and eggplant datasets, providing both sparse (eggplant only) and high coverage (yeast and eggplant) molecule sets. Mapping molecules to a reference is a more accurate process than mapping molecules to each other, since there is no stretching effect on the *in silico* reference map. Therefore, we formed a true alignment dataset by first mapping all molecules to the reference. From this dataset, we obtained a set of overlapping molecule pairs (>125Kb overlap length) and consider these to be the true molecule-to-molecule alignment pairs. High coverage molecule sets were formed by iteratively obtaining molecules which mapped to each yeast chromosome resp. to each eggplant contig longer than 5Mb. The median overlap rates of these molecule sets were 17 per molecule in eggplant and 113 per molecule in yeast. The sparse alignment dataset was produced only for eggplant data, by subsampling such that each molecule in the dataset is used in a single pairwise alignment.

**OptiMap performs better all-round.** Assessing all-versus-all pairwise alignment performance on the full molecule set as outlined above is computationally extremely intensive. We therefore randomly selected 5 contigs or chromosomes and selected only reads mapping on these within the high coverage molecule sets. OptiMap (dense mode) and RefAligner were then applied on all pairs in these sets. For OptiMap (dense mode), we used a score threshold of $S > 0.65$ for the naive mode (first round) and $S > 0.55$ for the sparse mode (second round). For RefAligner, we chose $10^{-10}$ for the maximum p-value threshold with the same minimum overlap length requirement as in the dense version of OptiMap (>125 kb). We iterated this overall procedure 20 times, where in each iteration we randomly draw 5 contigs/chromosomes with their corresponding molecules without replacement. The results (Fig 8A and 8B) showed that OptiMap's performance is better, in terms of both recall and precision. The eggplant alignment results (Fig 8B) indicate that dense mode increases the recall over the naive mode, without decreasing precision.

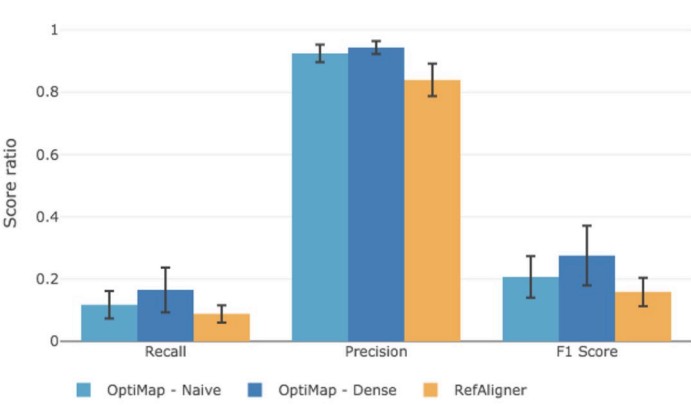

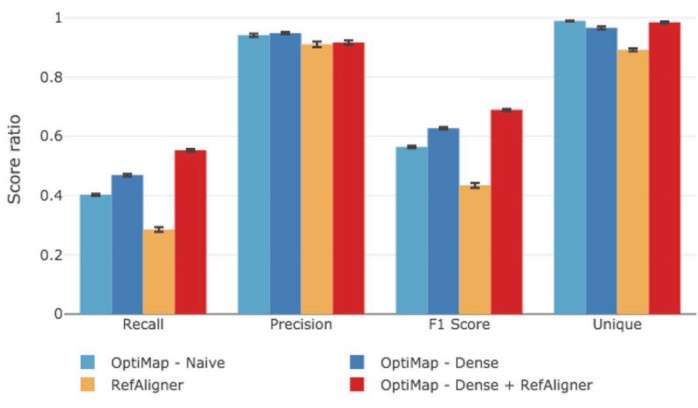

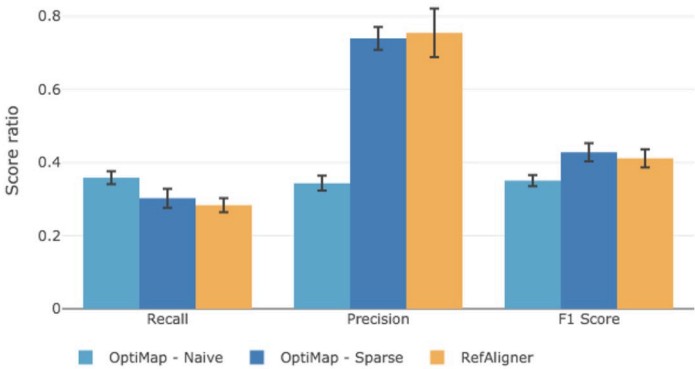

**Fig 8. Alignment performance for yeast (A) and eggplant data (B,C).** Precision, recall, unique molecules and F1 (harmonic mean) of alignments found for diferent molecule sets. A. Taken from all yeast contigs (avg. overlap rate 113x). B. Taken from the longest eggplant contigs (> 5Mb, avg. overlap rate 17x). C. Eggplant molecules with a single overlap.

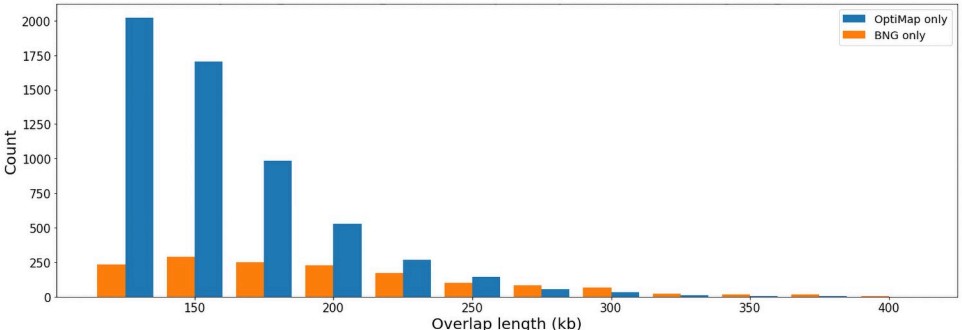

**Fig 9. Distribution of alignment overlap lengths for pairs aligned only by either of the tools.**

We then repeated the analysis with the sparse molecule set, this time using both OptiMap's naive and sparse modes. For both modes, we used default parameters and $S > 0.60$ as the score threshold. We report the results as precision, recall and the number of unique molecules found using the two methods in Fig 8C. OptiMap-sparse showed far better precision than OptiMap-naive. This is due to a combination of adding alignments with stretched molecules and discarding unspecific, false positive alignments based on too few labels (i.e. fewer than 9). Overall, this comes at the cost of a slightly decreased recall. BNG's Refaligner has an even slightly higher precision, but at a cost to recall which makes its overall F-score lower than that of OptiMap-sparse.

**OptiMap and RefAligner find different alignments.** Due to fundamental differences between the RefAligner and OptiMap approaches, either tends to find alignments which are not found by the other. Therefore, it could be beneficial to combine the alignment sets. To assess this, we calculated the performance for a combined set of alignments found by both OptiMap-dense and RefAligner (Fig 8B). As expected, this approach considerably increased the recall and the number of unique molecules found, reflecting the complementary nature of these two methods.

On the high coverage eggplant dataset, 56% of the correct alignments found by OptiMap-dense were not found by RefAligner. Many of these alignments had relatively shorter overlaps, as shown in Fig 9. OptiMap is considerably more sensitive than RefAligner in aligning molecules with shorter overlaps (<250kb), due to its use of intensity information. We show several examples of such alignments in Fig 10A, with clearly sufficient signal to capture a specific pattern with high precision. The increased precision at short overlap lengths can partly also be attributed to the alignments found after the transitive closure phase. We similarly inspected specific alignments only found by RefAligner (34% of the total number of RefAligner alignments) and visualized some in Fig 10B. It is apparent that DNA stretching pushed a large proportion of the signal peaks out of phase in the overlap, leading to lowering of scores. The stretching correction in OptiMap-sparse can only mitigate the effect of global stretching, while RefAligner's dynamic programming approach helps solve the local stretching prevalent in these cases. Interestingly, OptiMap-dense also found some long overlap alignments which could not be found by RefAligner. This shows that the score threshold drop made possible after the transitive closure helps capture partially out-of-phase alignments as well.

## OptiMap does not yet scale to large datasets

Construction of genome maps by optical mapping is dependent on a *de novo* assembly phase. The BNG toolset uses an overlap-layout-consensus (OLC) assembly approach, which finds all

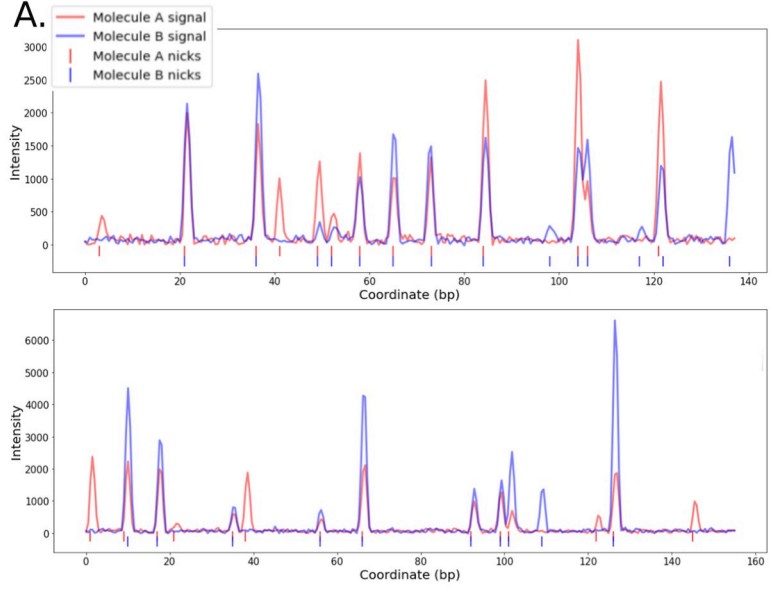

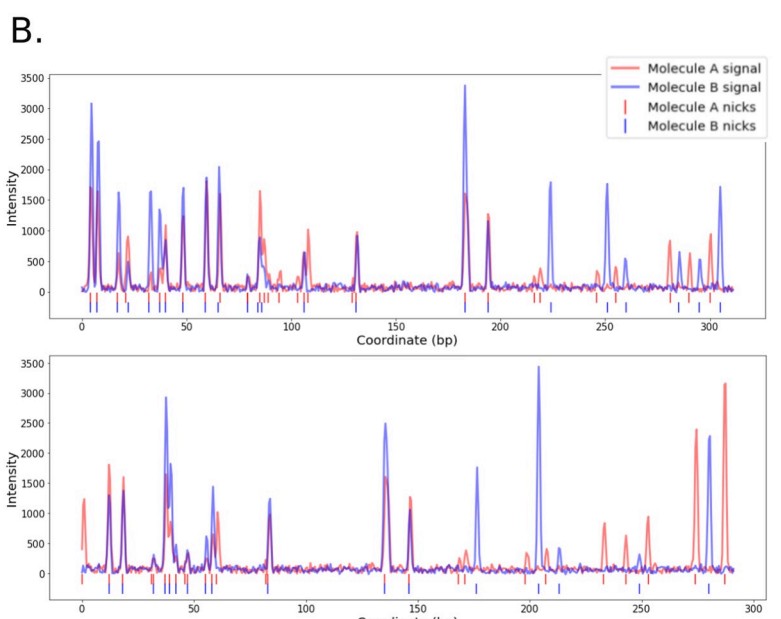

**Fig 10. Visualization of overlapping parts of molecule pairs aligned by A. OptiMap and B. RefAligner.** All overlaps are shown from OptiMap's perspective where only global stretching is applied. OptiMap alignment (A.) overlaps tend to be shorter in size, whereas RefAligner alignment overlaps indicate loss of phase due to local stretching after 200bp for the top and 150bp for the bottom alignment.

overlapping molecules with a pairwise all-versus-all alignment procedure to create edges in an OLC graph [17]. We assessed the use of OptiMap as an alternative to RefAligner for this alignment step. However, given to the all-vs-all pairwise normalized cross-correlation step required in OptiMap-naive, the first phase of OptiMap-dense and OptiMap-sparse, OptiMap has quadratic time complexity and does not scale to genome-size datasets for larger genomes. We extrapolated the computation time required for OptiMap-naive and found that it would take over 8,000 hours

to perform an all-vs-all alignment for 500k molecules (S1 Fig), which can be regarded as a lower bound for *de novo* assembly of a higher eukaryotic genome. This is clearly infeasible.

## Conclusion

High-throughput optical mapping technology is complementary to second and third generation sequencing and essential to achieve high-quality *de novo* reconstructions of complex genomes. Here, we provide a suite of tools to analyze, process and use the measurements generated by BioNano Genomics devices, currently the main platform to deliver such data. The tools start from the raw image data and take a novel signal-based approach to alignment, rather than translating these into sequences first as in existing methods.

OptiScan processes raw image frames to detect and extract molecule signals. Results showed that low-level improvements in this stage have a positive effect on downstream analyses such as mapping and assembly. Moreover, we showed that relative label peak heights contain information, which is lost when signals are translated to sequences of distances between peaks. Our signal-based alignment tool, OptiMap, finds alignments that show greater diversity of unique molecules and a higher number of total alignments when compared to BNG RefAligner. The main advantage of OptiMap is its ability to make more out of short but specific molecules. This can be partly attributed to the use of intensity information on label sites which increases the specificity of short matches. We further showed that due to the methodological differences between OptiMap and BNG's RefAligner tools, these two methods can be combined to achieve even better performance. This indicates that using OptiMap in the assembly process could achieve a higher final assembly depth. However, OptiMap is currently not as fast as RefAligner due to the pairwise cross-correlation step. In future we aim to address this with GPU-based parallelization and/or use of signal hashing methods to lower the time complexity.

For NGS sequence analysis—quality control, alignment, assembly, variation detection etc. —a plethora of open source tools are available. In contrast, for high-throughput optical mapping such tools are scarce and often critically depend on software provided by BNG. While this approach has proven successful in some applications of optical mapping [5, 11, 33, 34], it has thus far hampered diversification and implementation of novel methods for (comparative) genomics based on optical mapping data. Given its unique capability of generating contiguity information in the megabase range, optical mapping technology has great potential for use in fundamental and applied studies depending on long-range haplotyping, detection of structural variation and recombination, finding copy number variation etc. As an open source, extensible framework for OM analysis, alignment and assembly, OptiTools (OptiScan and OptiMap) increases the accessibility of OM data and offers a foundation for developing novel tools to address more high-level genomics challenges, such as optical map assembly. In addition, while both OptiScan and OptiMap can handle Saphyr data, their alignment performance have not been assessed yet on this type of data. Moreover, as results of individual tools can be exported to the standard BNG file formats—BNX for molecules and CMAP for alignments—users will be able to develop pipelines combining specific tools that best meet their needs.

## Supporting information

**S1 Fig. Estimated time required to compute of all-vs-all OptiMap-naive alignments.**
Obtained by fitting a quadratic function $f(x) = mx^2$ on run-times calculated for datasets consisting of 0 to 8,000 molecules with a step of 200. The optimal value of $m$, obtained using nonlinear least squares, was $1.34 \times 10^{-4}$ with a standard deviation of $8.8 \times 10^{-7}$.
(PDF)

## Acknowledgments

We thank Aude Darracq for testing software and Sander Peters, Linda Bakker, Geo Velikka-kam James, Nikkie van Bers and Glenda Willems for fruitful discussions.

## Author Contributions

**Conceptualization:** Mehmet Akdel, Henri van de Geest, Gabino Sanchez-Perez, Dick de Ridder.

**Data curation:** Mehmet Akdel, Henri van de Geest.

**Formal analysis:** Mehmet Akdel.

**Funding acquisition:** Gabino Sanchez-Perez, Dick de Ridder.

**Investigation:** Mehmet Akdel, Elio Schijlen.

**Methodology:** Mehmet Akdel.

**Project administration:** Dick de Ridder.

**Resources:** Irma M. H. van Rijswijck, Eddy J. Smid.

**Software:** Mehmet Akdel.

**Supervision:** Dick de Ridder.

**Visualization:** Mehmet Akdel.

**Writing – original draft:** Mehmet Akdel.

**Writing – review & editing:** Dick de Ridder.

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
