## [Decision Letter · Decision Letter 0]

2 Jul 2021

PONE-D-21-17430

Signal-based optical map alignment

PLOS ONE

Dear Dr. Mehmet Akdel,

Thank you for submitting your manuscript to PLOS ONE. After careful consideration, we feel that it has merit but does not fully meet PLOS ONE’s publication criteria as it currently stands. Therefore, we invite you to submit a revised version of the manuscript that addresses the points raised during the review process.

We look forward to receiving your revised manuscript.

Kind regards,

Jeong-Sun Seo

Academic Editor

PLOS ONE

Journal Requirements:

"This work is part of the Open Technology Programme with project number 14516,

(partly) financed by the Netherlands Organisation for Scientific Research (NWO),

domain Applied and Engineering Sciences, and supported by Bayer CropScience NV,

Genetwister Technologies BV, Rijk Zwaan BV and SESVanderHave NV. Yeast optical

map data was generated in a ZonMW Enabling Technologies Hotels project (435002001)."

"This work is supported by the research programme "High-throughput haplotyping using optical mapping" with grant number TTW 14516, which is financed by the Netherlands Organisation for Scientific Research (NWO, https://www.nwo.nl). The funders had no role in study design, data collection and analysis, decision to publish, or preparation of the manuscript."

Additionally, because some of your funding information pertains to commercial funding, we ask you to provide an updated Competing Interests statement, declaring all sources of commercial funding.

In your Competing Interests statement, please confirm that your commercial funding does not alter your adherence to PLOS ONE Editorial policies and criteria by including the following statement: "This does not alter our adherence to PLOS ONE policies on sharing data and materials.” as detailed online in our guide for authors  http://journals.plos.org/plosone/s/competing-interests.  If this statement is not true and your adherence to PLOS policies on sharing data and materials is altered, please explain how.

Please include the updated Competing Interests Statement and Funding Statement in your cover letter. We will change the online submission form on your behalf.

4. Please ensure that you refer to Figure 5 in your text as, if accepted, production will need this reference to link the reader to the figure.

Reviewers' comments:

Reviewer's Responses to Questions

**Comments to the Author**

1. Is the manuscript technically sound, and do the data support the conclusions?

Reviewer #1: Yes

2. Has the statistical analysis been performed appropriately and rigorously? 

Reviewer #1: Yes

3. Have the authors made all data underlying the findings in their manuscript fully available?

Reviewer #1: Yes

4. Is the manuscript presented in an intelligible fashion and written in standard English?

Reviewer #1: Yes

5. Review Comments to the Author

Reviewer #1: In this manuscript, authors have developed signal-based tools to extract (OptiScan) and analyze (OptiMap) data from raw image data generated with BioNano Genomics devices. These tools generally perform better than the softwares produced by the manufacturer. However, as the authors also have pointed out, OptiMap and RefAligner are methodologically different and can be combined to achieve better performance.

Although optical mapping has a not short history, the development of its analysis tool has not been much. Also, as this manuscript shows, there is a lot of room for analytical improvement. Therefore, this manuscript will contribute to broadening the scope of genomics research in accordance with this principle. But overall, there are some parts that are lacking in detailed explanation, and those are listed below.

1. Fig. 5 is not used in the text.

2. The authors used signal intensities to measure the real length between the peaks. Do the signal intensities from the peaks also reflect the number of cut sites in the pixel?

3. The alignment was iterated 20 times to compare OptiMap and RefAligner. Is this done on the same references with the same molecules?

4. In “For RefAligner, we chose 10 × 10^10 for the maximum p − value threshold”, What does this mean? How was p-value calculated for this?

5. What is (¿125 kb) in the “OptiMap performs better all-round” section?

6. In “ For OptiMap (dense mode), we used a score threshold of t = 0.65 for the naive mode (first round) and 0.55 for the sparse mode (second round)”, what does “t” mean?

7. Are the graphs in Fig 10 made with Optiscan or BGN (Are the data stretched)?

6. PLOS authors have the option to publish the peer review history of their article (what does this mean?). If published, this will include your full peer review and any attached files.

Reviewer #1: No

---

## [Author Response · Author response to Decision Letter 0]

23 Aug 2021

Dear Editor,

We thank the reviewer for their comments and suggestions. We believe the revisions made in response to these suggestions have improved the clarity of the manuscript. Please find below a point-by-point response to the comments. The changes are marked red in the manuscript.

* Reviewer's comment: Fig. 5 is not used in the text.

We thank the reviewer for picking this up. We intended to mention the figure in the text under the "Naive mode" and "Sparse overlap mode" sections. We have now added references to Fig. 5 (A, B and C) in the text (lines 199, 223, 226).

* Reviewer's comment: The authors used signal intensities to measure the real length between the peaks. Do the signal intensities from the peaks also reflect the number of cut sites in the pixel?

This is a good observation. We previously studied the relationship between peak height and the number of labels in the same pixel, however we could only find a low correlation. We trained a logistic regression model on a window centered on peaks to predict whether it contained single dual labels (one or two cuts/labels within a pixel). The model's precision was 63% at a recall of 43%. Some possible explanations of the low performance can be that the peak intensities have a high level of noise and/or there are more factors involved in peak height, such as surrounding sequence or local DNA structure. Further analysis is needed for a clearer view on this relationship. We have now added some sentences suggesting that further study is needed to look into what different patterns of peaks mean (lines 320-321).

* Reviewer's comment: The alignment was iterated 20 times to compare OptiMap and RefAligner. Is this done on the same references with the same molecules?

A random set of references (with corresponding molecules) as selected at each of 20 total iterations. We realise that this was not clear from the text. We have now changed the wording (lines 349-351).

* Reviewer's comment: In “For RefAligner, we chose 10 × 10^10 for the maximum p − value threshold”, What does this mean? How was p-value calculated for this?

We thank the reviewer for bringing this up. The score threshold had a typo and should have p < 10^-10; and we have now fixed this in the text (line 348). For the second part of the question, we could not find any literature or software manual on how the p was calculated by the Bionano Genomics software (RefAligner).

* Reviewer's comment: What is (¿125 kb) in the “OptiMap performs better all-round” section?

This was a typo and now changed to >125kb (line 349).

* Reviewer's comment: In “ For OptiMap (dense mode), we used a score threshold of t = 0.65 for the naive mode (first round) and 0.55 for the sparse mode (second round)”, what does “t” mean?

We tried to explain as score threshold in text here:

"We designate a pair of molecules as an aligning pair if the score S is above the threshold (t, default 0.60) and discard the remaining pairs."

However, we think that this was confusing and we replaced altogether and used the following notation: "S>0.65" throughout the paper to denote score threshold (lines 208, 229, 237, 243, 346, 347, 357).

* Reviewer's comment: Are the graphs in Fig 10 made with Optiscan or BGN (Are the data stretched)?

We thank the reviewer for their attention to the figure. The figures were made from the perspective of Optimap, after linear stretching, as it shows where Optimap goes wrong (Fig 10 B.: in the top example there is local stretching just after 200bp and in the bottom example after 150bp). This is now explained in the figure caption.

---

## [Decision Letter · Decision Letter 1]

16 Sep 2021

Signal-based optical map alignment

PONE-D-21-17430R1

Dear Dr. Mehmet Akdel,

We’re pleased to inform you that your manuscript has been judged scientifically suitable for publication and will be formally accepted for publication once it meets all outstanding technical requirements.

Kind regards,

Jeong-Sun Seo

Academic Editor

PLOS ONE

Reviewers' comments:

Reviewer's Responses to Questions

**Comments to the Author**

1. If the authors have adequately addressed your comments raised in a previous round of review and you feel that this manuscript is now acceptable for publication, you may indicate that here to bypass the “Comments to the Author” section, enter your conflict of interest statement in the “Confidential to Editor” section, and submit your "Accept" recommendation.

Reviewer #1: All comments have been addressed

2. Is the manuscript technically sound, and do the data support the conclusions?

Reviewer #1: Yes

3. Has the statistical analysis been performed appropriately and rigorously? 

Reviewer #1: Yes

4. Have the authors made all data underlying the findings in their manuscript fully available?

Reviewer #1: Yes

5. Is the manuscript presented in an intelligible fashion and written in standard English?

Reviewer #1: Yes

6. Review Comments to the Author

Reviewer #1: The authors have appropriately addressed all of my comments. Hence, I would like to recommend this manuscript to be accepted.

7. PLOS authors have the option to publish the peer review history of their article (what does this mean?). If published, this will include your full peer review and any attached files.

Reviewer #1: No

---

## [Editor Report · Acceptance letter]

22 Sep 2021

PONE-D-21-17430R1 

Signal-based optical map alignment 

Dear Dr. Akdel:

I'm pleased to inform you that your manuscript has been deemed suitable for publication in PLOS ONE. Congratulations! Your manuscript is now with our production department. 

Kind regards, 

on behalf of

Dr. Jeong-Sun Seo 

Academic Editor

PLOS ONE